# Structural and Dynamic Determinants of Molecular Recognition in Bile Acid-Binding Proteins

**DOI:** 10.3390/ijms23010505

**Published:** 2022-01-03

**Authors:** Orsolya Toke

**Affiliations:** Laboratory for NMR Spectroscopy, Structural Research Centre, Research Centre for Natural Sciences, 2 Magyar Tudósok Körútja, H-1117 Budapest, Hungary; toke.orsolya@ttk.hu; Tel.: +36-1-382-6575

**Keywords:** bile acids, enterohepatic circulation, protein structure, intracellular lipid binding proteins, ligand binding, positive cooperativity, site-selectivity, protein dynamics, conformational selection, fatty acid-binding proteins

## Abstract

Disorders in bile acid transport and metabolism have been related to a number of metabolic disease states, atherosclerosis, type-II diabetes, and cancer. Bile acid-binding proteins (BABPs), a subfamily of intracellular lipid-binding proteins (iLBPs), have a key role in the cellular trafficking and metabolic targeting of bile salts. Within the family of iLBPs, BABPs exhibit unique binding properties including positive binding cooperativity and site-selectivity, which in different tissues and organisms appears to be tailored to the local bile salt pool. Structural and biophysical studies of the past two decades have shed light on the mechanism of bile salt binding at the atomic level, providing us with a mechanistic picture of ligand entry and release, and the communication between the binding sites. In this review, we discuss the emerging view of bile salt recognition in intestinal- and liver-BABPs, with examples from both mammalian and non-mammalian species. The structural and dynamic determinants of the BABP-bile–salt interaction reviewed herein set the basis for the design and development of drug candidates targeting the transcellular traffic of bile salts in enterocytes and hepatocytes.

## 1. Introduction

Bile acids are amphipathic molecules synthesized from cholesterol in the liver [1], which, in the small intestine, facilitate the absorption of dietary lipids, cholesterol, and fat-soluble vitamins [2]. Besides their role in digestion, by interacting with receptors and activating signaling pathways, they participate in a diverse set of regulatory processes [3]. In particular, by the activation of the nuclear farnesoid receptor FXR α [4,5], they have a role in the regulation of their own synthesis, thereby, in cholesterol and whole-body lipid homeostasis [6], as well as in the control of glucose and energy metabolism [7]. Additionally, by being a ligand for the G-protein coupled receptor TGR5 [8,9] and activating mitogen-activated protein kinase pathways [10,11], they contribute to several additional cell-signaling and immunoregulatory processes. By affecting the absorption of electrolytes and induction of apoptosis, bile salts have also been found to influence intestinal and colonic epithelial function and integrity [12,13]. As a result of their diverse role in cellular processes and their cytotoxicity, disorders in bile acid transport and metabolism have been related to a number of metabolic diseases, atherosclerosis, type II diabetes, and to gastro-intestinal cancer development [7,14,15,16,17,18]. 

Bile acids consist of a sterol moiety with hydroxyl groups at specific positions, endowing the molecule with an amphipathic character necessary for the solubilization of lipids (Figure 1A).

Additionally, following their synthesis, the majority of bile acids undergo a glycine or taurine conjugation [26], giving rise to a negatively charged side chain. The most abundant bile salts in humans are glycocholic acid (GCA, 40%) and glycochenodeoxycholic acid (GCDA, 20%), with three (3α, 7α, 12α) and two (3α, 7α) hydroxyl-groups on the steroid rings, respectively, followed by the corresponding tauro-conjugated derivatives and secondary bile salts, which are formed in the intestine via dehydroxylation by bacterial enzymes [27,28]. The bile salt pool in different species differ substantially in terms of both the hydroxylation pattern of the sterol nucleus and the presence/absence and type of side-chain conjugation [29]. Moreover, changes in the composition of the bile salt pool are observed during human development and have also been associated with pathological conditions [30].

Following their synthesis, bile salts are secreted into the bile to facilitate the digestion of lipid-like compounds. From the intestine, they are efficiently transferred back to the liver via the portal vein, in a process termed enterohepatic circulation [14,31]. The recycling of bile salts involves multiple membrane crossing events and the transcellular trafficking of bile salts in enterocytes and hepatocytes. Crossing of the plasma membrane is accomplished by specific transporters [32] localized at the apical and basolateral membrane of enterocytes and hepatocytes, whereas the intracellular transport of bile salts is aided by bile acid-binding proteins (BABPs), a subfamily of intracellular lipid-binding proteins (iLBPs) [33,34]. Besides functioning as transcellular carriers of bile salts in enterocytes and hepatocytes, there is increasing evidence that BABPs have a role in gene expression by controlling the presentation of bile acids to FXR α [4] through a direct protein–protein interaction augmented by bile acids. Furthermore, the BABP expressed in the distal small intestine has been reported to be co-localized and functionally associated with the ileal bile acid transporter (IBAT), responsible for conjugated bile acid uptake in the ileum [35].

The family of iLBPs (14–16 kDa) evolved by successive gene duplications generating tissue specific homologues and, besides BABPs, include the subfamilies of fatty acid binding proteins (FABPs) and cellular retinol (retinoic acid) binding proteins (CR(A)BPs) [33,36]. Despite the relatively low sequence homology (eleven (forty) amino acid positions displaying identity (similarity), in over 75% of the analyzed 59 sequences, Appendix A), there is a high degree of topological conservation among iLBPs exhibiting a similar fold of a beta-barrel comprised of ten antiparallel beta-strands arranged into two orthologous beta-sheets covered by a helix-loop-helix motif at the top. Although the tertiary structure is essentially the same (Figure 1B), the structural and dynamic characteristics of molecular recognition in BABPs is fine-tuned to a system with more than one binding site, allowing a sophisticated regulation of the binding process according to the local bile salt pool. This has implications for not only bile salt trafficking and metabolic targeting, but also in buffering against a dangerously high level of cytotoxic bile salts in enterocytes and hepatocytes, as well as in possible ligand-induced protein–protein interactions involving BABPs. While FABPs [37] and CR(A)BPs [38] have been in the focus of interest for decades, a residue-level understanding of the mechanism of action of BABPs is only beginning to emerge. In this review, we discuss the latest findings regarding the structural and dynamic determinants of bile salt recognition in intestinal- and liver-BABPs, with examples from both mammalian- and non-mammalian species, compare the mechanistic picture of ligand binding with other members of the iLBP family, and explore the possible mechanisms of ligand transfer between BABPs and the plasma membrane.

## 2. Positive Binding Cooperativity

Similar to other iLBPs [33], the binding cavity (~1000 Å^3^) of BABPs is enclosed by two five-stranded antiparallel beta-sheets (Figure 1B) and a helix-loop-helix motif at the top [20,22,25,39,40]. Despite the common topology, unlike most other members of the iLBP family, such as FABPs and CR(A)BPs, which in most cases bind a single molecule of ligand [33], extensive studies of both human [41,42,43,44] and chicken [40,45,46] BABPs report a binding stoichiometry of 1:2. Additionally, besides the two internal binding sites, mass spectroscopic analysis and structural studies raise the possibility of the existence of additional weak binding site(s) on the protein exterior [20,47]. Mass spectrometric and calorimetric analysis of rabbit ileal BABP also show a binding stoichiometry of 1:3 [48], whereas in zebrafish liver BABP, the presence or absence of a disulphide bridge have been found to affect the stoichiometry [49].

Bile salt binding in BABPs exhibits a varying degree of positive cooperativity depending on the hydroxylation pattern of the steroid moiety of the bound bile salts. This has been shown elegantly for the human analogue, where the binding thermodynamics of bile salts with different hydroxylation pattern and conjugation has been analyzed systematically [50]. Independent of the presence/absence and type of conjugation, bile salts with three hydroxyl groups on the steroid rings (i.e., cholates and its conjugated derivatives) have been found to exhibit the highest degree of cooperativity with Hill coefficients of 1.8–1.9. Remarkably, the contribution of positive cooperativity to the Gibbs-free energy of GCA binding has been shown to exceed the intrinsic affinity of the binding sites [41]. In agreement with the calorimetric measurements, a stopped-flow kinetic study of bile salt binding in human I-BABP revealed a multistep binding mechanism with kinetic parameters reflecting a high and moderate degree of positive cooperativity for the binding of GCA and GCDA, respectively [44]. The same kinetic studies provide evidence of conformational transitions in the binding process. Specifically, a rate-limiting unimolecular step on the ms time scale has been associated with a transition between a closed and a more open *apo* state regulating ligand entry, whereas a conformational change on the time scale of seconds occurring after the second binding step has been found to be related to the positive cooperativity of binding [44].

The level of positive cooperativity in BABPs varies in different species and tissues. For example, when comparing the free energy of interaction between the two binding sites in different intestinal BABP analogues, the binding of GCDA exhibits the largest level of positive cooperativity in the human form [41], followed by the rabbit [48] and zebrafish [20] analogues, and finally by the chicken protein, where NMR chemical shift perturbations indicate a structural rearrangement following the first binding event [45]. When seeking for the structural determinants of energetic coupling between the binding sites, mutagenesis and NMR structural work on the thermodynamically most stable heterotypic human I-BABP:GCDA:GCA complex has revealed the existence of a hydrogen-bond network between the two binding sites with the participation of amino acid side chains and the hydroxyl-groups of bound bile salts [43]. Specifically, an upper network involving the steroid ring hydroxyl group at position C-12, and a lower network involving the steroid ring hydroxyl groups at positions C-3 and C-7, have been identified with side-chain contributions from W49, Q51, N61 (upper), and Q99, S101, E110, R121 (lower) (Figure 2). Remarkably, a gain of function approach in the poorly cooperative chicken ileal form has succeeded in turning the wild-type analogue into a cooperative double mutant (H99Q/A101S) by replacing residues at two key positions [51] with amino acids present in the human protein, allowing the formation of a continuous pattern of hydrophilic interactions anchoring the ligands.

Besides species-dependence, binding cooperativity shows tissue-dependence as well. This is highlighted by the difference between the liver and ileal BABP analogues in chicken. Unlike the poorly cooperative ileal form, the liver analogue binds GCDA with a high degree of positive cooperativity [52]. Intriguingly, the strength of cooperativity of GCDA binding in the chicken liver form is significantly higher than observed for the human ileal protein. Noteworthy, the binding thermodynamics of GCDA in cL-BABP is more comparable to that of the highly cooperative GCA in hI-BABP, suggesting that the structural and dynamic determinants of binding cooperativity in different species and tissues are likely tailored according to the local bile salt pool. Consistent with positive binding cooperativity, NMR lineshape analysis in cL-BABP reveals an initial fast-exchange process with an intermediate of low population representing the singly ligated species, followed by a second step with a low k*_off_*, corresponding to the high-affinity binding of the second ligand molecule [46]. Investigations of the chicken liver analogue highlights the role of H98 in the binding process [40]. Specifically, pH-dependent measurements suggest that the deprotonated form of H98, by interacting with the proximate I111 (Figure 2B), may have a role in establishing a hydrophobic bridge between the bound bile salts, thereby enhancing the communication between the two sites [53]. Additionally, mutation H98Q in cL-BABP has been found to diminish positive binding cooperativity by making the second binding step less favorable [52]. By elucidating the diffusion coefficient of ligand species, the authors have also shown that the affinity loss in the mutant primarily arises from one of the two binding sites. A similar loss of positive cooperativity is observed in the human ileal form by mutation Q99A, highlighting the involvement of the hydrogen bond donor/acceptor side chains at the beginning of βH in the energetic communication [43]. Additionally, as shown for the human form, the thermodynamic coupling between the binding sites is affected only in the case of the binding of GCDA, but not of GCA, indicating different pathways of communication for di- and trihydroxy bile salts in the mutant.

## 3. Site Preference of Bile Salts

Positive binding cooperativity is accompanied by a site-preference of di- and trihydroxy bile salts in several BABPs including both mammalian and non-mammalian analogues. As demonstrated by NMR spectroscopic measurements on complexes of human I-BABP with mixtures of isotopically labeled/unlabeled bile salts [42], unlabeled GCA displaces ^15^N-labeled GCDA from one of the binding sites, whereas unlabeled GCDA displaces ^15^N-labeled GCA from the other binding site. Given the nearly identical intrinsic affinities of GCA for site 1 and 2 [41], initially it was hypothesized that site-selectivity in the heterotypic GCDA/GCA complex arises from differences in cooperativity, i.e., that GCA, the trihydroxy bile salt, can exert its high cooperative effect through site 2 only. However, subsequent isothermal titration calorimetric studies of functionally impaired human I-BABP mutants have shown no correlation between losses in site-selectivity and losses in macroscopic binding cooperativity, indicating that the two phenomena are not linked as closely as previously thought [43]. Structural and dynamic investigations of human I-BABP mutant Q51A lacking site-selectivity suggests that the strong intrinsic affinity of GCDA for site 1 is more likely to have a role in the site preference of bile salts in the heterotypic human I-BABP:GCDA:GCA complex [54]. This is consistent with MD simulations, indicating that by allowing a better closure of the helical cap, a deep penetration of the more hydrophobic dihydroxy bile salt (i.e., GCDA) into the protein core at site 1 could have a role in site-selectivity [47].

In the chicken liver BABP analogue, the site-selectivity of bile salts appears to be related to the presence/absence of a disulphide bridge [55]. Specifically, in the T91C polymorph, the presence of a disulphide bridge (C80-C91) linking βF and βG, while leaving binding stoichiometry and the overall binding mechanism unchanged, introduces a site preference of GCA and GCDA. Although the site-selectivity is not exclusive, the presence of the S-S bridge is able to revert the relative populations of homo- and heterotypic complexes [53]. Additionally, besides site-selectivity, the disulphide-containing and disulphide-free chicken L-BABP variants also differ in binding cooperativity with the site-selective T91C form exhibiting a weaker coupling between the binding sites [46]. There is a further indication in the chicken liver analogue that cooperativity and site-selectivity are not directly linked in BABPs. The authors have also shown that the introduction of the C80-C91 S-S bridge results in a change of side chain conformation for three key residues in the H/I protein region (E99, Q100, E109) [53], allowing the formation of an extended intermolecular H-bond network and stabilization of the side chain of H98, a residue that anchors GCA to one of the sites [40]. Importantly, while backbone conformational changes associated with the formation of the disulphide bridge in cL-BABP are small, significant changes in backbone motions distributed over the entire protein are observed in both the *apo* and the *holo* forms [55], suggesting that differences in the propagation of dynamic fluctuations may be associated with site-selectivity in the protein.

## 4. Conformational Changes of the Protein Backbone upon Bile Salt Binding

The analysis of *apo* and *holo* structures of BABPs [25,39,40,56] and other iLBPs [57,58,59,60,61] reveals four main protein regions, which show a major structural rearrangement upon ligand binding: the E-F region, the C/D- and G/H-turns, and the helical cap (in particular, α-II) (Figure 3). We note that at the moment, only the coordinates of the heterotypic (GCDA:GCA) complex is available for the human form [25], whereas the chicken liver analogue is a homotypic complex with CDA [40]. In the chicken liver form, comparable displacements in the Cα trace occur in additional turn regions such as the B/C-, D/E-, and F/G-turns, as well. While the overall pattern of conformational rearrangement upon ligand binding within the iLBP family is similar, the observed differences could be related to differences in binding stoichiometry (i.e., BABPs vs. FABPs/CR(A)BPs) and the position and stabilizing interactions of ligand(s) in the binding pocket (*cf below*).

In both the human ileal and the chicken liver BABP-analogues, the EF-region shows the most significant reorganization of the protein backbone. Upon ligation, four (five) new H-bonds are formed between βE and βF in hI-BABP (cL-BABP), accompanied by a displacement of the EF-hairpin toward βD. The increase in backbone stability in the *holo* form is reflected in increased protection from solvent exchange in hI-BABP, as observed for a near continuous segment of residues between T73-A81 [62]. We note that neither in the *apo* or in the *holo* forms are any hydrogen-bonds between the backbone atoms of βD and βE limiting the source of stabilization to van der Waals contacts between hydrophobic side chains and making the EF-region pliable to conformational fluctuations. In the chicken liver analogue, there is a large displacement of βG as well upon ligation [40], lacking any stabilizing H-bonds with the adjacent βF and βH strands in the empty form [56]. In contrast, in hI-BABP, βG is fairly well ordered in the *apo* state as well [39], and the conformational rearrangement upon bile salt binding in the region is restricted to the G/H-turn and the beginning of βH [25]. Intriguingly, the displacement of the G/H-turn in the two analogues are opposite in the sense that in the human ileal form, it moves away from the EF-region, whereas in the chicken liver protein, it moves toward the EF-hairpin. Importantly, the structural rearrangement in the GH-region accompanying ligand binding results in increased stabilization for the adjacent βI in both BABPs.

**Figure 3 ijms-23-00505-f003:**
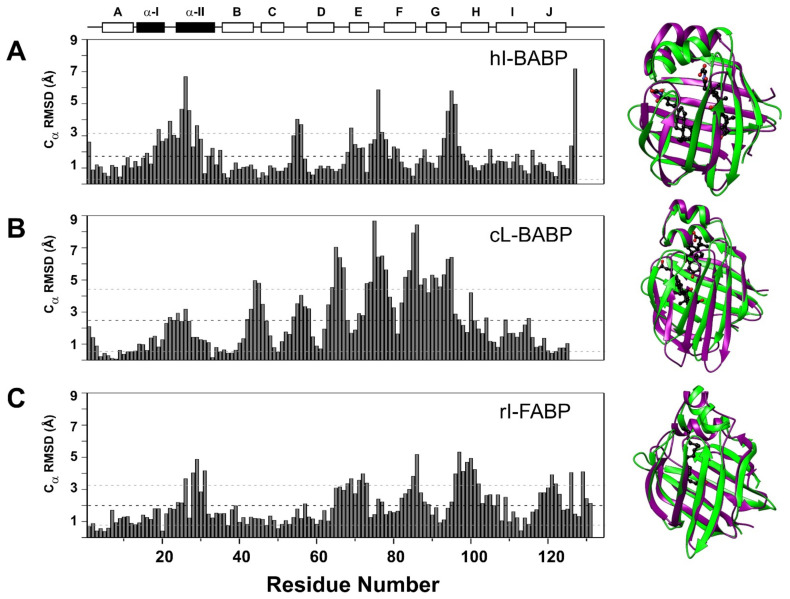
Backbone conformational change upon ligand binding. Displacement calculated for the C_α_ trace between (**A**) the *apo* (PDB: 1o1u [39]) and the heterotypic doubly ligated (hI-BABP:GCA:GCDA, PDB: 2mm3 [25]) forms of human ileal BABP, (**B**) the *apo* (PDB: 1zry [56]) and the homotypic doubly ligated (cL-BABP:CDA:CDA, PDB: 2jn3 [40]) forms of chicken liver BABP, and (**C**) the *apo* (PDB: 1ael [63]) and *holo* (PDB: 1ure [59]) forms of rat intestinal FABP. Secondary structure elements are indicated at the top. Superimposed ribbon diagrams of the corresponding structures are shown on the right with the *apo* and *holo* forms in magenta and green, respectively. Bound ligands are in black, with oxygen atoms shown in red.

The third region showing the most significant change upon ligand binding in BABPs is the C/D-turn, which moves closer to the helical cap and the E/F-turn region upon ligand binding. We note that in the chicken liver analogue, residues ^52^HYSGGHT^58^ are replaced by ^52^KTPRQTV^58^, endowing the turn with substantially more rigidity in both the *apo* and the *holo* forms. An important difference between hI-BABP and cL-BABP is that in the latter, the helical region is much less affected by bile salt binding. Specifically, while in cL-BABP, the Cα displacement in α-II averages about 1.4 ± 0.9 Å (with max. at D26 of 3.2 Å), in the human ileal form it is 3.2 ± 1.8 Å (with max. at D26 of 6.8 Å) manifested in a ~15° change in orientation of the helix upon binding, covering the binding cavity. Importantly, unlike in FABPs and CR(A)BPs, where the helical region displays a high degree of backbone disorder in the *apo* state [61,63,64], in BABPs, it appears to be well defined in both the *apo* and the *holo* states.

## 5. Binding Cavity in BABP Complexes

An unusual, conserved feature of the internal binding cavity of iLBPs is its polar face. In the case of BABPs, H-bond donor/acceptor side chains provide a major source of stabilization for the bound bile salts with multiple OH-groups on their steroid rings. Besides anchoring the ligands, in cases of positive binding cooperativity, as exemplified by the doubly ligated heterotypic hI-BABP:GCDA:GCA complex (PDB code: 2mm3) [25], H-bonds provide a way of communication between the binding sites as well (Figure 4A). A key determinant of bile salt binding in hI-BABP is W49 (βC) located in the middle of the binding pocket. In addition to forming extensive hydrophobic contacts with the steroid rings of both bile salts, the NH-group of its indole side chain forms a H-bond with the 12α-OH of the trihydoxy bile salt (GCA) at site 2. In addition to W49, GCA is stabilized by the interaction of its 7α-OH with the proximate S101 (βH) and E110 (βI) side chains as part of a H-bond network, additionally involving the backbone atoms of R121 (βJ). Besides the interactions with the bound ligands, W49 forms two H-bonds with the backbone heteroatoms of N61 (βD), which in turn is involved in H-bonds with Q51 (βC) and T73 (βE). The latter, further stabilized by a H-bond with K77, is within H-bond distance to the carboxylate side chain of GCDA at site 1. Additional residues involved in anchoring the dihydroxy bile salt to site 1 in hI-BABP are Q99 and N61 by interactions with the 3α- and 7α-OH groups of its steroid ring system, respectively. In addition to H-bonds, van der Waals interactions with nearby aliphatic and aromatic side chains provide an important source of stabilization for the ligands. With the exception of W49, they are primarily involved in forming the wall of the binding pocket creating a favorable environment for the hydrophobic face of the bound bile salts.

When comparing the coordinates of the bound ligands in hI-BABP:GCDA:GCA with the superimposed structure of the available homotypic complex of the chicken ileal analogue (PDB code: 2lba) [51], the coordinates of GCDA at site 1 are nearly superimposable. However, the GCDA molecule at site 2 in the cI-BABP-complex is located significantly closer to the helical cap, with its side chain wormed in between α-II and the C/D-turn (Figure 4B). While the interaction pattern around site 1 is fairly similar in the two analogues, the pathway of communication to the bile salt trapped in the upper segment of the binding pocket is missing in the chicken analogue, resulting in the lack of positive cooperativity. Noteworthy, by introducing a H-bond donor/acceptor side chain in the vicinity of the 3α-OH of GCDA at site 1 and loosening up of the Y97-H99-E110-R121 H-bond network in double mutant H99Q/A101S, cooperativity could be introduced into the chicken ileal analogue as well [51]. Similar to cI-BABP, the bile salt at site 2 appears to be trapped in the upper part of the binding pocket in the chicken liver analogue (PDB code: 2jn3) [40] as well (Figure 4C). The difference in the position of one of the ligand molecules might arise from the different orientation of α-II in the two analogues. In hI-BABP, α-II changes its orientation substantially upon ligand binding and by interactions with residues in the C/D- (Y53) and E/F-turns (M74) contributes significantly to the stability of the *holo* state. In the chicken variants, α-II is oriented more upward in both the *apo* and the *holo* states, forming less contact with the two turn regions (Figure 3). Another key difference between hI-BABP and cL-BABP is the substitution of W49 by a valine in the latter. This has a profound effect on the interaction network stabilizing the OH-groups of the steroid rings, which becomes limited to residues of βH (H98, Q100), βI (E109), and βJ (R120) in the vicinity of site 1 [40]. An additional source of stabilization of bound bile salts in each analogue is provided by positively charged residues in the vicinity of the carboxylate groups of bile salt side chains. In the chicken forms (both liver and ileal), such stabilization is contributed by lysine/arginine residues in both the C/D- and the E/F-turns, whereas in the human ileal form by K77 of the E/F-turn only. An additional source of stabilization for the carboxylate group of bile salt at site 2 in hI-BABP may arise through the tautomer equilibrium of H57 in the C/D-turn, whose pK_a_ increases markedly upon ligand binding [65].

When comparing the position of ligands in doubly ligated BABPs and singly ligated FABPs/CR(A)BPs (Figure 5), it occurs that the single fatty acid/retinol/retinoic acid ligand in the latter is shifted to approximately the middle of the binding pocket, and is located roughly in between the two bound bile salts when superimposed [57,58,59,61,66,67,68,69]. Unlike in BABPs, the major source of stabilization for the bound fatty acid/retinoic acid is provided by interactions between the alkyl chain of the ligand and residues with bulky hydrophobic side chains of the proximate turn regions (in particular C/D and E/F), some of the strands (βD, βH, βI, βJ), and the helical cap. The carboxylate group of the bound fatty/retinoic acid interacts with polar side chains and ordered solvent molecules in the cavity. The latter appears to be a conserved feature of iLBPs, including BABPs [57,68,70,71,72,73]. Importantly, the ordered water molecules and their coordinating side-chain atoms are often located in nearly the same positions in the absence and the presence of ligands. They have been suggested to be an integral part of the structural framework of the binding cavity, with a role in the regulation of the conformational space of both the cavity and the ligand [60]. From a comparative study of pig ileal BABP and a bovine heart FABP, it appears that the cavity of BABP exhibits a considerably faster solvent exchange [73], which might be related to the fact that the energetic communication between the two binding sites in BABPs requires a certain degree of conformational flexibility. This is consistent with the notion that amino acid sequences could have evolved not only for the optimization of interactions with a specific ligand type, but also for the optimization of the arrangement and dynamics of internal water [73].

## 6. Backbone Mobility on the Fast Time Scale

Because of their strong effect on entropy, motions on the picosecond to nanosecond time scale have high biological importance. Similar to other iLBPs, the *apo* state of BABPs displays an elevated level of fast internal mobility in specific segments such as the EF-region, the C/D-, and some other turn-regions (e.g., the F/G-turn in cL-BABP, the I/J-turn in hI-BABP) [56,62]. Regarding the helical region, while in cL-BABP α-II exhibits an increased degree of flexibility, in hI-BABP, it remains rigid on the ps-ns timescale. Overall, as indicated by order parameters of N-H bond vectors inferred from NMR relaxation analysis [74], BABPs are substantially more rigid on the ps-ns time scale than most FABPs [63,75,76] and even CR(A)BPs [61,64,69]. Additionally, in comparison with the two other subfamilies, bile salt binding is accompanied by only a modest stiffening of the protein backbone [56,62]. This is most noticeable when comparing ligand-induced changes in ps-ns flexibility in the helical cap and the proximate C/D- and E/F-turns (Figure 5). However, while the differences are numerically smaller, stiffening appears to be more extensive in hI-BABP, encompassing the most part of the N-terminal beta-sheet (in particular, βB, βC, βD) and a proximate segment of βJ, together with residues in the E/F-hairpin. Further studies of backbone dynamics in hI-BABP suggest different responses to temperature increase in the *apo* and the *holo* forms, indicating differences in local conformational heat capacities between the two ligation states [77].

In comparison to FABPs, the dynamics of the helical region differs even more in cL-BABP, where ps motions remain persisting in the bound state as well [40]. NMR measurements agree with MD simulation data, showing that root-mean-square fluctuations of the protein backbone decrease substantially for the C/D- and E/F-turns, but increase for the helix-loop-helix region upon bile salt binding [78]. The authors attribute the increased motional freedom of the helices in *holo* cL-BABP to a missing H-bond between α-I and βJ, resulting in less constraint between the beta-barrel and the helical region. Taken together, the dynamics of the helical region in BABPs differ significantly from what was observed in FABPs, which (i) exhibit substantially more disorder in α-II in the *apo* state, and (ii) by interactions with the proximate C/D-turn, ligand binding shifts the order–disorder equilibrium toward an ordered, closed state [60,63]. The absence of these features suggests that in BABPs, the helical region is likely to have a less direct role in mediating ligand exchange than in FABPs and CR(A)BPs.

The more rigid nature of the helical region in BABPs is likely related to the observed differences in the stability of the helix-less variants between BABPs and FABPs. Specifically, while the Δ17-SG I-FABP-variant lacking the helical cap exhibits only a modest decrease in stability leaving the folding-unfolding transition reversible and highly cooperative [79], a helix-less variant of rabbit I-BABP displays a dramatically reduced melting point, high disorder, and the loss of cooperative transition in the *apo* form [48]. Remarkably, similar to rat I-FABP, the helix-less variant of rabbit I-BABP still has the capability of ligand binding, which enhances its stability and has been shown to re-establish the cooperativity of unfolding in the protein. We note that regarding the helical region, an interesting distinction exists between CRBP-I and FABPs as well, with α-I appearing to be more mobile than α-II in *apo* CRBP-I [61]. Nevertheless, similar to the ordering of α-II in FABPs upon fatty acid binding [63], a stiffening of α-I (without the change of the mobility of α-II) is evoked in CRBP-I upon retinol binding.

While in FABPs and CRBPs ps-ns flexibility is primarily associated with the accessibility of the otherwise enclosed binding cavity and ligand exchange, there are indications that internal motions on the fast time scale in BABPs can affect bile salt binding in a more specific manner as well. This has been suggested for cL-BABP, where the presence or absence of a disulphide bridge between two adjacent beta strands altering the rigidity of the protein backbone has been found to be able to modulate the site-preference of ligands [55]. In a more recent study of the human ileal form, functionally impairing mutations deteriorating site-selectivity (Q51A) or positive binding cooperativity (Q99A) have been found to enhance the ps-ns flexibility of several key residues in the binding pocket [54].

## 7. Conformational Fluctuations on the Millisecond Timescale

Slight perturbations in fast dynamics induced by ligand binding in BABPs are accompanied by major changes in slow motions. In the *apo* state of both ileal and liver BABP proteins, CPMG-based NMR spin relaxation analysis directed at the investigation of slow dynamics [80] reports an extensive network of conformational fluctuations on the fast end of ms time scale, which ceases upon bile salt binding [56,62,81]. Relaxation dispersion (R_ex_) profiles of individual residues in hI-BABP fit well to a global two-state exchange process between a major ground and a low populated (~1–5%) higher energy state (Figure 6A) [62,77]. Importantly, the time scale of the observed conformational fluctuation in the *apo* state matches the time scale of a rate-limiting conformational change preceding ligand binding in BABPs and FABPs inferred from stopped-flow kinetic measurements [44,82]. In both chicken and human BABP analogues, the most intense slow internal fluctuations are observed in the C-terminal face of the β-barrel, primarily in the EFGH-region, which in hI-BABP extends toward β-strands I and J as well (Figure 6B). Additionally, in hI-BABP, there is a second group of residues undergoing motion with a slightly slower exchange rate. The slower cluster is confined mainly to the N-terminal half of the protein involving α-I, βB with the preceding linker to α-II, and the C/D-turn region continuing in βD. Temperature-dependent NMR relaxation measurements show an entropy–enthalpy compensation for both clusters characteristic of disorder–order transitions [77] with energy barrier between the exchanging states of ~20 kcal/mol (Figure 6B). Supporting the hypothesis of exchange between the ground and a less ordered higher energy state, a joint analysis of R_ex_ measurements and NMR thermal melting data [83] indicates a structural and energetic connection between ms timescale conformational fluctuations at and below room temperature and thermal unfolding (60 °C) of hI-BABP. Consistent with the NMR data, MD simulations show evidence of correlated motions involving the helical cap, the E-F strands, and the bottom of the β-barrel. The NMR measurements further suggest that the higher energy state shows similarities with a more open, partially unfolded state exhibiting a native-like residual structure in the N-terminal half and more disordered segments in the EFGH region [83]. According to the analysis, the more open state has a high susceptibility to dimerization which, as indicated by MD simulations, is most likely due to the exposure of hydrophobic patches. The formation of transient aggregates as a consequence of conformational fluctuations has also been suggested for cL-BABP with a subset of residues in regions of high plasticity exhibiting a singular behavior upon urea unfolding [84]. The exposure of hydrophobic residues upon functionally related conformational fluctuations appears to be a general characteristic of iLBPs posing an aggregation risk in the absence of ligands. The balance between protein function, conformational plasticity, and aggregation vulnerability has been investigated in detail in CRABP-I [85,86], highlighting the role of protecting mechanisms, which despite the risk of aggregation, enables proteins to retain conformational flexibility to function while avoiding unacceptable aggregation propensity [87,88].

The ms-timescale dynamics in both ileal [65] and liver BABPs [56] is thought to be associated with the protonation equilibria of histidines, providing a pH-dependent allosteric regulation of a transition between a closed and a more open protein state. In cL-BABP, the observed conformational fluctuation has been related to the tautomer equilibrium of a buried histidine (H98) located near the G/H turn. As indicated by pH-dependent NMR measurements and MD simulations, the protonation of H98 triggers a marked change in both the observed μs-ms dynamics and the conformation of the E/F-loop region at the open side of the β-barrel [56]. According to the simulations and chemical shift perturbation data, propagation of fluctuations from H98 to the E/F-turn is mediated by a chain of buried polar residues. The key role of H98 in cL-BABP dynamics has also been indicated by mutagenesis, with substitution H98Q attenuating the μs-ms timescale mobility of the protein [52].

Strikingly, while H98 is conserved in the human ileal analogue as well, there is a significant difference between the two proteins in terms of both the pK_a_ of H98 and its change upon bile salt binding. Unlike in the chicken liver form, where H98 is buried and exhibits a pK_a_ of 4.7 in the *apo* state, which upon bile salt binding shifts further down [40], in hI-BABP it is facing the solvent in both ligation states with values of pK_a_ of 6.6 (*apo*) and 7.0 (*holo*) [65]. Accordingly, while in the chicken liver analogue the contribution of H98 toward substrate binding is favored by an increase in pH, in the human ileal form it is the opposite. The latter is supported by a pH-dependent analysis of the thermodynamics and kinetics of bile salt binding in hI-BABP showing an increase in the overall binding affinity and the association rate constant of the first binding step below the pK_a_ of H98 [65]. Importantly, in hI-BABP there are two additional histidines in the C/D-turn as well, which similar to the E/F- and G/H-regions undergoes a conformational change upon ligand binding. Most importantly, there is a near complete overlap between residues exhibiting a change in chemical environment upon pH change and those exhibiting a conformational exchange confirming the link between protonation/deprotonation of histidines and the observed global conformational fluctuation in the human ileal analogue as well. Importantly, similar to ligand binding, lowering the pH below the pK_a_ of histidines results in a near complete cessation of ms motion throughout the protein (Figure 6C) indicating a similarity in the dynamic behavior of the histidine-protonated *apo* form and the complexed state of hI-BABP. On the contrary, increasing the pH above the pK_a_ of histidines results in an increase in the exchange rate and the population of the higher energy state. Hence, conformational exchange in hI-BABP is likely to be triggered by the formation of the deprotonated state. Of note, there is a difference in the dynamic response to pH between the clusters of the N- and C-terminal half of the protein with the BCD-αIαII region exhibiting a steeper increase in the exchange rate upon the elevation of pH than the EFGH-region. Taken together, while the overall mechanism of protonation-evoked opening-closing transition in cL- and hI-BABPs is similar, with the presence of histidines in both the C/D- and the G/H-turn regions, the human ileal form appears to exhibit a more complex regulation of the conformational equilibrium.

The advantage of NMR relaxation dispersion measurements is that in addition to the kinetic and thermodynamic parameters of the exchange process, structural information on the low-populated otherwise ‘invisible’ higher energy state can be obtained [80]. Specifically, residue-specific chemical shift differences between the ground (G) and higher energy (E) states (|Δω_EG_|) inferred from the measurements are (i) indicative of protein regions undergoing the largest amplitude conformational motion and (ii) provide us with information about the nature of the higher energy state. In both ileal [65,77] and liver [46] BABP analogues, the largest values of |Δω_EG_| (2–3 ppm) are detected in the E/F- and G/H-turn regions and in βH (Figure 6B). Additionally, in hI-BABP, residues in the C/D-turn region, in βD, βB, and in the link between α-II and βB display chemical shift differences > 1 ppm. When mapped on the ribbon diagram of the *apo*/*holo* states, these regions match the ones which undergo the most significant conformational change upon bile salt binding. Moreover, in both chicken liver and human ileal BABP, there is a subset of residues (not involved in direct contact with the ligands in the bound form), which show a correlation between |Δω_EG_| and the chemical shift difference between the *apo* and the *holo* forms suggesting that the higher energy state has a conformation reminiscent of the bound state. This strongly indicates that by visiting conformational states in specific protein regions corresponding to the *holo* form, the observed global fluctuation(s) in unligated BABPs has a role in mediating bile salt binding. We note that similarly to the human analogue, in porcine I-BABP, although no global fit has been performed, a significant portion of residues distributed throughout the protein show indication of μs-ms timescale conformational motion, which silences upon binding [81].

Importantly, the observed slow dynamics is not limited to BABPs but have a general functional role in iLBPs. This is strongly suggested by NMR spectroscopic investigations of FABPs [63,75] and CR(A)BPs [64,89] showing evidence of the contribution of conformational exchange to transverse relaxation in specific protein regions. Noteworthy, analysis of the dynamic behaviour of different iLBP subfamilies [90] shows distinctions in the affected regions suggesting alterations in the regulation of ligand entry mechanisms (*cf below*). A unique example is the human liver fatty acid-binding protein, which has been found to bind a variety of hydrophobic/amphipathic lipid-like compounds (e.g., oleate, fatty acid CaA thioesters, lysophosphatic acid, bile salts) and similarly to the human ileal and chicken liver BABPs, shows the involvement of a large number of residues in conformational exchange [91]. However, unlike in hI-BABP, exchange in the human liver analogue displays a high degree of heterogeneity suggesting that the protein may visit multiple higher energy states. Intriguingly, unlike in hI-BABP or cL-BABP, slow motions in the human liver analogue are retained or even enhanced upon binding, underlying the plasticity of the ligated form, which the authors associate with the promiscuity of ligand recognition [91]. 

## 8. Ligand Entry and Release Mechanisms in BABPs and Other iLBPs

Ligand uptake and targeted release are fundamental functional requirements for carrier proteins such as iLBPs. Comparison of the X-ray structures of *apo*/*holo* pairs of various iLBPs shows the lack of obvious openings large enough for the ligand to enter and exit from the enclosed binding site. However, a thorough inspection of the X-ray structure of rat I-FABP in the absence and presence of fatty acid reveals a relatively low density of atom packing in a region bounded by α-II and the C/D- and E/F-turns and became the first indication of a likely location for a ligand entry portal [57]. Originally it was hypothesized that ligand binding is accomplished by the displacement of ordered water molecules without major conformational changes in the protein [33,58]. Subsequent NMR studies and stopped-flow fluorescence measurements on full-length and the helix-less variant of rat I-FABP revealed a rate-limiting kinetic step in ligand binding, which the authors interpreted as a conformational rearrangement involving the helical region [79,82]. Analysis of the helix-less variant was followed by a detailed NMR structural and dynamic investigation of the protein with and without bound fatty acids leading to the so-called dynamic portal hypothesis [60,63]. According to this, an order–disorder equilibrium exists in the *apo* state involving a localized region of backbone disorder in α-II and the neighboring C/D-turn. Upon ligand binding, by a series of stabilizing interactions reminiscent of a helix capping box, the equilibrium is shifted toward the more ordered, closed state [60]. In agreement with the dynamic portal hypothesis, relaxation analysis of the side chain dynamics in the human I-FABP analogue reveals reduced mobility of several methyl groups upon oleic acid binding in the portal region [92]. Besides the NMR works, binding study of a triple mutant (V32G/F57G/K58G) FABP of murine adipocyte with an enlarged portal region provides additional support for the portal hypothesis in FABPs [93]. We note that in addition to α-II, the subsequent linker to βB, and the C/D-turn, stiffening of the protein backbone upon ligand binding in rat I-FABP, is also indicated in the E/F-turn and, to a lesser extent, in βJ [63]. Similar to FABPs, NMR spectroscopic investigations [61] and MD simulations [94] of CRBP-I and CRBP-II show a decreased flexibility of the protein backbone upon retinol binding in α-II, and the C/D- and E/F-turns indicating a more general applicability of the portal hypothesis in iLBPs. Importantly, stiffening of the backbone in specific regions upon ligand binding on the ps-ns timescale is accompanied by a suppression of conformational fluctuations on the μs-ms timescale as well [64]. Of note, while silencing of slow motions is evident in both CRBP-I and CRBP-II upon retinol binding, a significant heterogeneity is observed in the exchange rates suggesting that instead of a simple two-state exchange, the higher energy state is likely to represent a manifold of disordered states. Moreover, the observed dynamics appears to be significantly faster in CRBP-II, in particular in specific regions such as α-II and the C/D-turn, suggesting that the two homologues may differ in the way they mediate ligand entry and release and interaction with the cell membrane [64]. The importance of the region bounded by the helical cap and the C/D- and E/F-turns in facilitating ligand entry and release in the subfamily of CRBPs is further indicated by a study of a double-mutant (A35C/T57C) of CRABP-I, in which ligand-induced repositioning of the helices and the two proximate turn regions has been shown to trigger the formation of a disulphide bridge (absent in the *apo* form), which in turn results in the ceasing of structural fluctuations and locking the ligand inside the binding cavity [95].

Similar to CRBPs, BABPs appear to be substantially more rigid on the ps-ns timescale than FABPs. Moreover, upon bile salt binding only a small decrease in flexibility is detected on this timescale. The major, more dominant form of internal flexibility in BABPs is an extensive network of μs-ms fluctuations in the *apo* form silenced upon ligand binding [56,62,81]. Remarkably, conformational exchange in BABPs does not display a heterogeneity found in CRBPs but fit well assuming a global two-state exchange throughout the protein (Figure 6). Additionally, remarkable is the overlap between regions undergoing a conformational change upon bile salt binding and the regions involved in the conformational fluctuation in the unligated form [46,77]. Moreover, the correlation of the ‘dynamic’ chemical shift difference between the ground and the higher energy state inferred from the CPMG relaxation dispersion measurements with the chemical shift difference between the *apo* and *holo* forms as observed for a group of residues in both hI-BABP and cL-BABP is a strong indication of the functional relevance of motion and that the excited state corresponds to a *holo*-like conformation visited by the *apo* state. Based on these observations, a conformational selection mechanism of ligand entry has been proposed for both the chicken liver [56] and human ileal [25] BABP analogues. According to this, an equilibrium between a state 1 and a state 2 is envisioned, which is shifted upon ligand binding (Figure 7). State 1, characterized by a closed EFGH-region and an open helical cap, is thermodynamically favored in the absence of ligands. State 2, exhibiting an enlarged gap between the E/F- and G/H-turn regions provides a corridor between the bulk solvent and the binding cavity and is thermodynamically favored in the presence of bile salts. In hI-BABP, opening of the EFGH-region is accompanied by the closure of the helical cap and repositioning of the C/D-turn toward the C-terminal half of α-II providing an additional source of stabilization. In cL-BABP with no significant conformational exchange in the N-terminal half, the contribution of the helical region appears to be less significant. The *EFGH-open* and the *EFGH-open/CDα-closed* state in the presence of bile salts in cL- and hI-BABP, respectively, is stabilized by numerous local and long-range interactions such as extension of H-bonding networks and hydrophobic interactions [25,40].

## 9. Ligand Transfer between BABPs and the Cell Membrane

Enterohepatic circulation facilitated by BABPs involves an intracellular transport of bile salts between the apical and basolateral membrane of enterocytes and hepatocytes. While the mechanism of bile salt uptake through the membrane is not yet fully clear, the most accepted view currently is that a lower pH and dielectric constant in the vicinity of the membrane may shift the conformational equilibrium of BABPs toward a more open state assisting bile salt entry. For instance, hI-BABP has been shown to be co-localized and functionally associated with the ileal bile acid transporter [35], which in turn interacts with the vacuolar H^+^-ATPase at the apical membrane [96]. Given the histidine protonation/deprotonation mediated shift in the conformational equilibrium between the closed and more open portal region in BABPs, fluctuations in pH brought about by the proton pump could have a mediatory role in bile salt uptake.

Intriguingly, while hI-BABP associates only weakly with lipid bilayers [97], cL-BABP binds to negatively charged phospholipid vesicles [98,99,100]. According to FTIR spectroscopy, interaction with the lipid vesicles is accompanied by a conformational change of the protein involving a decrease in secondary structure [98]. Importantly, pH-dependent measurements unveil the presence of partly folded states under acidic condition suggesting that membrane binding may occur via a molten globule-like state [101]. The conformational transition can be reversed by increasing the salt concentration indicating that the interaction is driven by electrostatic forces and remains to be peripheral without penetration into the hydrophobic acyl chains of the membrane [98]. It has also been suggested that depending on the phase state of the lipid membrane, different degrees of partial unfolding may occur [102]. As indicated by stopped-flow kinetic measurements, the interaction with the membrane can be described by a three-state kinetic model involving a fast step of binding of cL-BABP to the anionic lipid headgroups with a simultaneous partial unfolding followed by a slower kinetic step upon which the monitored tryptophan (W6) in βA releases contacts with the aqueous phase [103]. As shown by X-ray diffraction methodologies, association of cL-BABP with the lipid vesicles affects the membrane architecture by increasing the interlamellar spacing and decreasing the compactness of the lipids [98]. A residue-level insight into the role of lipid-membrane interaction in bile salt uptake has been obtained from an NMR spectroscopic analysis of the tripartite cL-BABP/bile salt/membrane system showing that lipid vesicles with anionic lipid headgroups and bile salts establish competing binding equilibria [99]. As shown by the authors, addition of bile salt to the protein-liposome mixture (with a partially unfolded protein conformation) shifts the equilibrium toward the liposome-free fully folded *holo* state. A specific protein region interacting with the liposomes has been elucidated corresponding to a patch at the bottom of the β-barrel [99,100]. It has to be mentioned that, as pointed out by MD simulations (30 ns) of the cL-BABP-membrane interaction, bile salt transfer may not necessarily occur through the protein region that is in direct contact with the membrane [104]. According to the simulations, the largest conformational change upon membrane association has in fact occur in α-II, the C/D-turn, and the E/F-region, corresponding to the BABP portal region. Remarkably, these regions correspond to segments displaying the largest fluctuation in secondary structure already in the membrane-free *apo* state as well. In a more recent study of hI-BABP, MD simulations on a longer timescale (1 μs) have identified the bottom of the beta-barrel as a together-moving region with the E/F hairpin [83] consistent with the dynamic coupling between the portal and the bottom of the barrel.

According to the most accepted model, the vectorial transport of bile salts in the cytosol of hepatocytes is governed by the difference in bile salt concentration in the proximity of the basolateral and the apical membrane, which by shifting the equilibrium between the *holo*/membrane-free and *apo*/membrane-bound state could have a regulatory role [99]. Nevertheless, the observed direct contact of cL-BABP with negatively charged phospholipid bilayers shows similarities with that described for the transfer of fatty acids between rat heart FABP [105] as well as various other (intestinal, keratinocyte, adipocyte) rodent and human FABPs [106] and the cell membrane, where ligand is released after a direct collision of the protein with the membrane. This suggests that bile acid release and transfer from cL-BABPs to the cell membrane might occur by a mixed mechanism of intrinsic and membrane-induced effects, with the latter augmenting the inherently existing conformational equilibrium between a closed and a more open protein state.

The interaction with the lipid bilayer in iLBPs is thought to be driven by electrostatic forces between positively charged lysine/arginine residues on the protein surface and negatively charged lipid headgroups [107,108]. While in FABPs, positively charged side chains in the helical- and the C/D-turn regions [109] together with the amphipathic character of α-I [110] have been shown to have a key role in membrane recognition, in cL-BABP two lysine residues (K43, K103) at the bottom of the beta-barrel are implicated in membrane binding with the lysine in α-I having no effect on membrane binding affinity [100]. We note that while the common isoform of the human ileal BABP analogue present in enterocytes associates with lipid vesicles only very weakly hindering the detailed structural analysis of the interaction, its long variant with a 25-residue extension at the N-terminus overexpressed in colorectal cancer exhibits similar cL-BABP membrane binding behavior [111]. Although the biological implication of the extension is not yet fully clear, the five additional Lys/Arg residues in the long variant significantly enhance the positively charged surface potential of the protein. In particular, the presence of the extension affects the chemical environment of backbone amides at the bottom of the β-barrel [111] where the epitope of membrane binding in the chicken liver variant has been identified [99]. In the common isoform of hI-BABP, the two implicated lysines are replaced by an aspartate and a valine hence lacking the necessary electrostatic driving force for the interaction.

## 10. Concluding Remarks

Being responsible for the transcellular trafficking of bile salts in enterocytes and hepatocytes, BABPs have a role in a diverse set of metabolic and regulatory processes. Their functioning as bile salt carriers is regulated by a combination of positive binding cooperativity and site selectivity, providing an efficient mechanism for the recognition of structurally diverse bile salts dictated by the local bile salt pool. As reviewed herein, structural and biophysical studies of the past two decades have provided a mechanistic view of the binding process at the residue or even atomic level setting the basis for the development of small molecule ligands for the modulation of BABP-regulated bile salt trafficking and for the engineering of BABP-based molecular carriers.

In addition to deciphering the determinants of cooperativity and site selectivity in different BABP analogues, a considerable effort has been devoted to the elucidation of ligand uptake and release mechanisms. In both the chicken liver and the human ileal forms, the two most extensively studied BABPs, a conformational selection mechanism of bile salt entry has been proposed with a pre-existing equilibrium in the *apo* state shifted toward a more closed conformation upon ligand binding. While internal dynamic processes on the ms timescale have been described for other iLBPs as well, a pH-dependent regulation of the binding process has so far only been evidenced for BABPs. Although the protonation deprotonation equilibrium of histidine(s) has a major role in conformational exchange in both cL- and hI-BABP, solvent exposure of histidines appears to markedly affect the pH range, where fluctuations aiding ligand entry are maximized. The pH-dependent conformational exchange between a closed and a more open state of BABPs has implications for bile salt uptake and release to the cell membrane during enterohepatic circulation. According to the most accepted view of the binding process, fluctuations in pH brought about by proton pumps in the vicinity of the membrane are thought to mediate a protonation/deprotonation triggered shift between a closed and a more open BABP conformation. While an atomic-scale picture of the membrane interaction is missing, electrostatic interactions are likely the driving force for a peripheral association of BABPs with the membrane. Solid state NMR measurements planned for the future by several groups are expected to provide us with a more detailed picture of the BABP-membrane interaction. An important difference between the chicken liver and the human ileal forms is the order of magnitude lower membrane affinity of the latter. Intriguingly, the strength of the interaction increases significantly in a long variant of hI-BABP overexpressed in colorectal cancer, the biological implications of which remains to be explored.

An important attribute of BABP-membrane interaction is a partial unfolding of the protein reversed upon ligand binding. Remarkably, partial unfolding in the membrane bound state affects the same protein regions undergoing a ms timescale conformational fluctuation in the membrane-free *apo* form related to the protonation/deprotonation equilibrium of histidines underlining the role of conformational exchange in ligand transfer. Importantly, partial unfolding processes in BABPs associated with ligand uptake and release could also have implications for ligand-mediated nuclear translocation mechanisms and the stimulation of nuclear receptors such as FXR α. In two related proteins, CRABP-II [112] and FABP4 [113], nuclear localization and nuclear export signals have been shown to mediate nuclear translocation, and it has been suggested to be a general mechanism of signaling in the family of iLBPs. Regions of enhanced plasticity (i.e., the DEF β-strands, linker connecting the helical cap to the β-barrel) observed in hI-BABP [83] and cL-BABP [84] could have a role in exposing the corresponding hydrophobic and basic residues required for signaling. Furthermore, the reported co-localization of hI-BABP with FXR α [35] in the nucleus raises the possibility of a direct interaction between the two proteins in a manner similar to that shown for CRABP-II and the retinoic acid receptor (RAR) [114] and hypothesized for FABPs and peroxisomal proliferator-activated receptors (PPARs) [115]. Accordingly, the role of BABPs in ligand-mediated nuclear receptor activation remains to be one of the most exciting questions for the future. Achievements of the past two decades in the understanding of BABP-bile–salt interaction reviewed herein shall provide a basis for such new directions in the field.

## Figures and Tables

**Figure 1 ijms-23-00505-f001:**
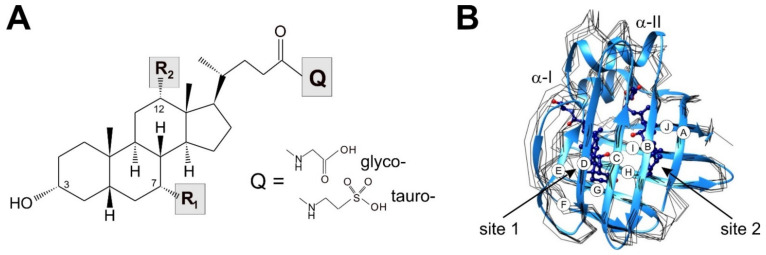
(**A**) Chemical structure of physiologically relevant bile acids differing in the hydroxylation pattern of the steroid rings (R_1_ = OH, R_2_ = OH cholic acid; R_1_ = OH, R_2_ = H chenodeoxycholic acid; R_1_ = H, R_2_ = OH deoxycholic acid) and the type of side-chain conjugation (glycine, taurine). (**B**) Backbone traces (black) of superimposed crystal structures of six BABPs (*apo* form) available in the PDB database (human ileal, PDB: 5l8i [19]; zebrafish ileal, PDB: 3elx [20]; human liver, PDB: 3stn [21]; chicken liver, PDB: 1tvq [22] and 7o0j [23]; toad liver, PDB: 1p6p [24]). In cases when more than one molecule is in the asymmetric unit, chain A is shown. For comparison, the ribbon diagram of the most representative element of the NMR-derived lowest-energy structural ensemble of *holo* human I-BABP (PDB: 2mm3 [25], blue) is shown with bound GCDA and GCA at site 1 and 2, respectively. β-strands are labeled A-J for identification when discussed in the text.

**Figure 2 ijms-23-00505-f002:**
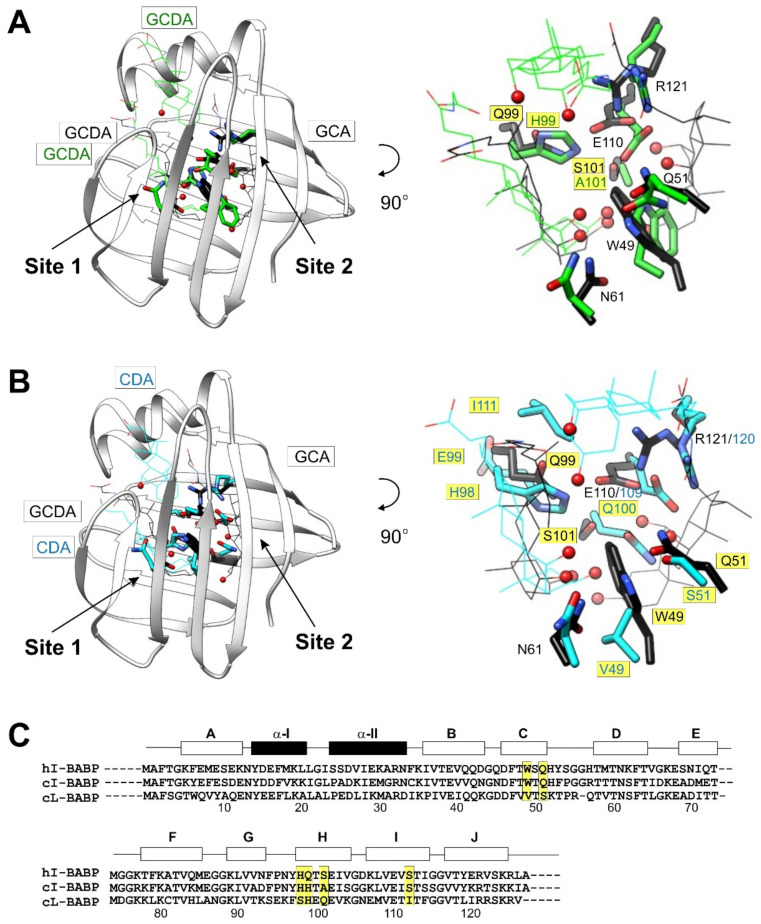
Key determinants of positive binding cooperativity and site-selectivity discussed in the text. Superimposed view of the binding pocket of human I-BABP:GCDA:GCA (PDB: 2mm3 [25], black) and (**A**) chicken I-BABP:GCDA:GCDA (PDB: 2lba [51], green) and (**B**) chicken L-BABP:CDA:CDA (PDB: 2jn3 [40], cyan) highlighting the key hydrogen-bond donor/acceptor residues (sticks) anchoring the ligands (wire) and providing a communication between the two binding sites. Amino acid replacements in comparison to hI-BABP in the two other complexes are marked in yellow. For clarity, only the ribbon diagram of the human complex is shown (left). Steroid ring OH-groups of the bound bile salts are shown as red balls. (**C**) Sequence alignment of the three BABP analogues shown in (**A**,**B**). Key amino acid replacements with implications for binding cooperativity and site-selectivity are highlighted in yellow. Note the missing glycine residue at position 56 in cL-BABP, affecting the numbering in panel (**B**). Secondary structure elements are indicated at the top. A more inclusive sequence alignment of BABPs and iLBPs are shown in Appendix A.

**Figure 4 ijms-23-00505-f004:**
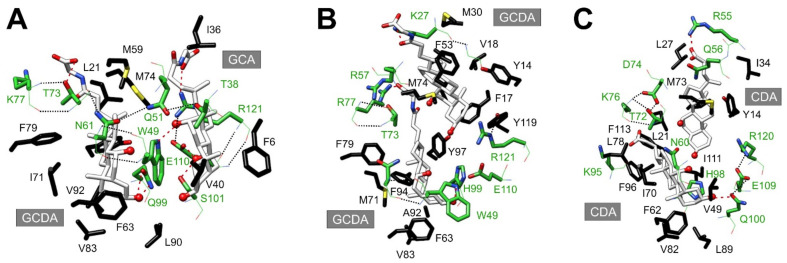
Stabilizing H-bond and hydrophobic interactions in the binding pocket of (**A**) human ileal (PDB: 2mm3 [25]), (**B**) chicken ileal (PDB: 2lba [51]), and (**C**) chicken liver (PDB: 2jn3 [40]) BABPs. Backbone and side chain atoms of residues with a polar side chain (green) are shown as lines and sticks, respectively. Hydrophobic amino acid side chains interacting with the apolar face of the bound bile salts are shown in black. Intermolecular (red) and intramolecular (black) H-bonds are indicated by dotted lines. Steroid ring OH-groups of bound bile salts are shown as red balls. Color coding: C-atoms in green (protein, polar residues), black (protein, apolar residues), or white (bile salts), O-atoms in red, N atoms in blue, S-atoms in yellow.

**Figure 5 ijms-23-00505-f005:**
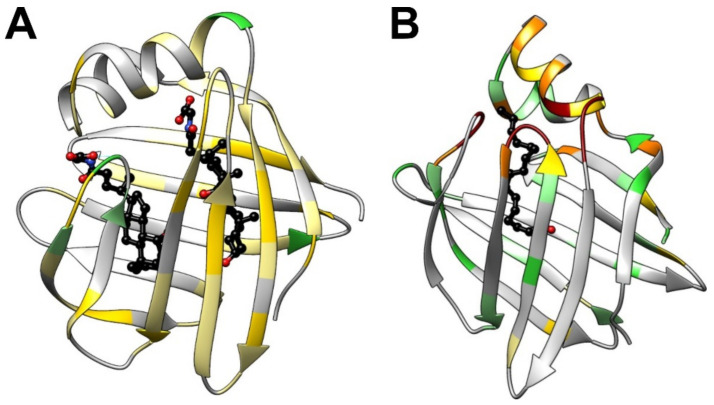
Differences in the generalized order parameter (S^2^) characteristic of the amplitude of the ps-ns motion of backbone amide NH bond vectors upon ligand binding mapped on the ribbon diagram of (**A**) the doubly ligated complex of human ileal BABP (hI-BABP:GCA:GCDA, PDB: 2mm3 [25]) and (**B**) the complex of rat intestinal FABP with palmitate (PDB: 2ure [59]). Color coding: 0.5 < S^2^*_holo_-*S^2^*_apo_* in red, 0.25 < S^2^*_holo_-*S^2^*_apo_* < 0.5 in orange, 0.1 < S^2^*_holo_-*S^2^*_apo_* < 0.25 in gold, 0.05 < S^2^*_holo_-*S^2^*_apo_* < 0.1 in khaki, −0.1 < S^2^*_holo_-*S^2^*_apo_* < 0.05 in pale green, S^2^*_holo_-*S^2^*_apo_* < −0.1 in green.

**Figure 6 ijms-23-00505-f006:**
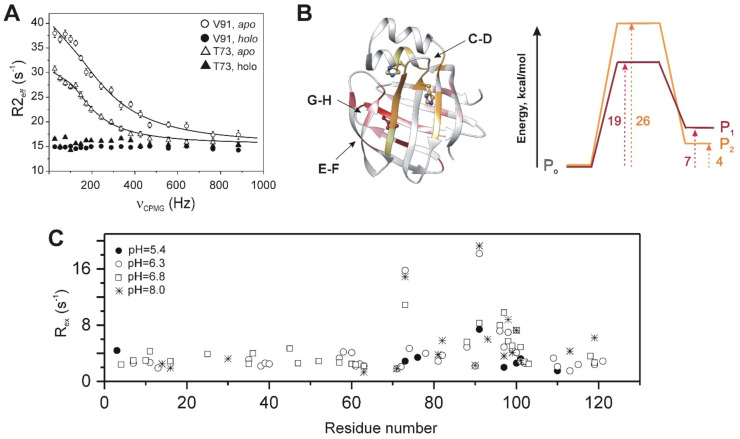
Slow dynamics in human ileal BABP mediating ligand entry as determined by ^15^N NMR relaxation dispersion measurements. (**A**) Contribution from conformational exchange (R_ex_) to transverse relaxation of backbone ^15^N nuclei of T73 (triangles) and V91 (circles) in *apo* hI-BABP (empty symbols) and the heterotypic hI-BABP:GCA:GCDA complex (filled symbols) as a function of the CPMG field strength. Ceasing of μs-ms-timescale motions upon bile salt binding is indicated by the flat R_ex_ profile in the *holo* form. Solid lines represent best-fit curves corresponding to a global fit of 24 residues assuming a two-state exchange process with an exchange rate constant, k_ex_ = 836 ± 59 s^−1^ and population of the higher energy state, p_e_ = 3.1 ± 0.2% (10 °C). (**B**) *Left:* Residues sensing a conformational exchange in the temperature range of 10–18 °C in *apo* hI-BABP can be grouped into two clusters with slightly different exchange rates. Chemical shift differences between the ground and the scarcely populated higher energy state deduced from NMR relaxation measurements are mapped on the ribbon diagram (PDB: 1o1u [39]) in pink-to-red and yellow-to-orange gradient for the faster and the slower cluster, respectively. The three histidines (H52, H57, H98) are highlighted in a ball-and-stick representation. Profiles showing the energy barrier and the free energy difference between the ground and the higher energy state for the faster (P_o_ ↔ P_1_, red) and the slower (P_o_ ↔ P_2_, orange) conformational fluctuation in *apo* hI-BABP deduced from temperature-dependent CPMG relaxation dispersion measurements are shown on the right. (**C**) Contribution from conformational exchange to transverse relaxation in *apo* hI-BABP at different values of pH. Note the considerably smaller number of residues (filled symbols), sensing a conformational fluctuation below the pK_a_ (6.4–6.6 as determined by NMR) of histidines.

**Figure 7 ijms-23-00505-f007:**
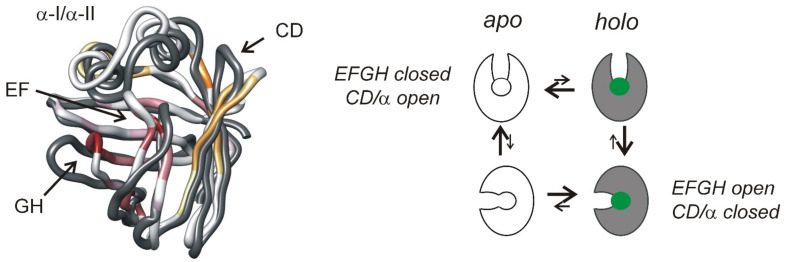
Proposed mechanism of bile salt entry. Superimposed backbone trace of the *apo* (PDB: 1o1u [39], white with color) and the *holo* form (PDB: 2mm3 [25], grey) of hI-BABP showing the enlarged gap between the E/F and G/H turns in the bound state connecting the binding cavity with the bulk phase. Note the accompanying closing of the helical cap upon bile salt binding. The color coding in the *apo* state reflects the chemical shift difference between the ground and the low-populated higher energy state inferred from the CPMG relaxation dispersion measurements (Figure 6) for the two clusters (orange and red) at pH = 6.3 with darker color corresponding to larger values of |Δω_EG_|. Note the match between regions undergoing a large conformational rearrangement upon bile salt binding and the darker shades of orange and red reflecting a larger chemical shift difference between the exchanging states in the *apo* form. Cartoon of the conformational selection model of ligand binding is shown on the right. A conformational equilibrium between an EFGH*^closed^*-CD/α*^open^* and an EFGH*^open^*-CD/α*^closed^* state in the *apo* form is shifted toward the latter upon bile salt binding.

## Data Availability

Not applicable.

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
