# Peer review of "Structural and Dynamic Determinants of Molecular Recognition in Bile Acid-Binding Proteins"

_ijms, 2022, doi:10.3390/ijms23010505_

Round 1

Reviewer 1 Report

The author reviews the achievements of the last two decades dealing with a residue-level understanding of the mechanism of action of bile acid binding proteins (BABPs). The findings regarding the structural and dynamic determinants of bile salt recognition in intestinal- and liver-BABPs from both mammalian- and non-mammalian species, are discussed. Common mechanisms of ligand binding and release and transfer between BABPs and the plasma membrane emerge from the comparative analysis.

The review is an excellent contribution to the field as the author identifies common binding features of BABPs from mammalian and non-mammalian species. Previously elusive connections are clearly stated allowing to propose a common binding and release mechanism of bile salts. We are convinced that this review will serve the novices as a detailed introduction to the field as well as the experts, alike to receive input about new directions of the field.

We only have minor suggestions, which we leave to the discretion of the author:

1) We believe that a swap of the order of sections 5 and 6 would increase the manuscript readability. Both section 5 and 7 describe protein dynamics, on two different time scales, in both apo and holo states. The structural details of protein/ligand interactions (section 6), could be discussed before starting with the description of the protein dynamics features.

2) The comparative analysis of human ileal and chicken liver BABP dynamics highlights  a common behaviour where “the overlap between regions undergoing a conformational change upon bile salt binding and the regions involved in the conformational fluctuation in the unligated form” and support a  conformational selection mechanism of ligand entry for both proteins. These results are in some extent summarized for hI-BABP in Figure 5D. To increase the manuscript readability we would suggest to move and summarize these results in a dedicated figure to be inserted in paragraph 8. The color/shape code of the of the conformational selection model cartoon needs to be better explained in the legend. The “label EFGH open/CD-alpha closed” (close to the bottom left state) should be moved to the bottom right state.

Minor remarks:

Page 1, line 18 “set” instead of “sets”

Page 3, line 80: “relatively low sequence homology”. It could be useful to associate a number deduced from the alignment reported in the Supplementary Material

Page 6, Beta strands should be labelled with the same format throughout the manuscript as bA, bB, …. (see for example line 206 or 236)

Page 11, line 393: Citation of Figure 3 would increase the comprehension of the last sentence

Page 13, line 455: T73 instead of Thr73, the same for V91

Page 19, line 704: “the extension”

Page 20, lines 740-743. Rewrite the sentence to clarify.

Page 21, line 767: “sequence” instead of “sequential”

Author Response

We thank the reviewer for his or her valuable comments and suggestions. Please find below our responses:

1) We believe that a swap of the order of sections 5 and 6 would increase the manuscript readability. Both section 5 and 7 describe protein dynamics, on two different time scales, in both apo and holo states. The structural details of protein/ligand interactions (section 6), could be discussed before starting with the description of the protein dynamics features.

Response

We thank the reviewer for bringing this into our attention. We have changed the order of sections 5 and 6 in the revision as it has been suggested.

2) The comparative analysis of human ileal and chicken liver BABP dynamics highlights  a common behaviour where “the overlap between regions undergoing a conformational change upon bile salt binding and the regions involved in the conformational fluctuation in the unligated form” and support a  conformational selection mechanism of ligand entry for both proteins. These results are in some extent summarized for hI-BABP in Figure 5D. To increase the manuscript readability we would suggest to move and summarize these results in a dedicated figure to be inserted in paragraph 8. The color/shape code of the of the conformational selection model cartoon needs to be better explained in the legend. The “label EFGH open/CD-alpha closed” (close to the bottom left state) should be moved to the bottom right state.

Response

We agree with the reviewer and moved the corresponding panel of the figure into section 8. (Figure 7 in the revised manuscript.) We have explained the colour coding in more detail now. The label the reviewer is referring to has been moved to the right side of the cartoon.

Minor remarks:

Page 1, line 18 “set” instead of “sets”

Response: It has been fixed.

Page 3, line 80: “relatively low sequence homology”. It could be useful to associate a number deduced from the alignment reported in the Supplementary Material

Response:  The number of amino acids displaying identity and similarity in over 75% of the analysed sequences has been given in the revised version.

Page 6, Beta strands should be labelled with the same format throughout the manuscript as bA, bB, …. (see for example line 206 or 236)

Response:  We have changed the labelling to be consistent throughout the manscript.

Page 11, line 393: Citation of Figure 3 would increase the comprehension of the last sentence

Response:  We have included a citation to the corresponding figure.

Page 13, line 455: T73 instead of Thr73, the same for V91

Response: It has been fixed.

Page 19, line 704: “the extension”

Response: It has been fixed.

Page 20, lines 740-743. Rewrite the sentence to clarify.

Response:  We have rephrased the sentence (pg. 767-770 in the revised version).

Page 21, line 767: “sequence” instead of “sequential”.

Response:  It has been fixed.

Reviewer 2 Report

The manuscript entitled “Structural and Dynamic Determinants of Molecular Recognition in Bile Acid-Binding Proteins” is centered on bile acid-binding proteins (BABPs), a subfamily of intracellular lipid-binding proteins (iLBPs), having a key role in cellular trafficking and metabolic targeting of bile salts. The manuscript focuses on the mechanism of bile salt binding, showing a mechanistic picture of ligand entry and release and of the communication between the binding sites, defined through structural and biophysical studies. I would recommend manuscript publication after addressing the following minor issues.

Minor issues:

  1. Only 13 works over the 111 cited in the manuscript have been published in the last 5-years (2016-2021), and only 27 over 111 in last decade (2011-2021).
  2. Figure 1B. The five BABPs used for the structure alignment should be reported in the caption (further their PBD codes). Further the structural model 1tvq, a recent high-resolution crystal structure of clBABP has been reported (PDB code 7O0J).
  3. Citing references should be added after each PDB code throughout the manuscript (and figure captions).
  4. Line 52. The hyphen after glycine seems not necessary and could be removed.
  5. The description of the binding cavity in BABP complexes, reported in section 6, could be moved before the actual section 2, describing the positive binding cooperativity. A preliminary description of the main features of the binding cavity could facilitate the understanding of the binding cooperativity (section 2), the site preferences (section 3), and the conformational changes occurring upon bile salts binding (section 4). Furthermore, various protein residues reported in these sections are shown in Figure 4, associated with section 6. Introducing Figure 4 before these descriptions would help the reader to have a better idea of the positions of the residues mentioned during the discussion. Otherwise, a figure showing the binding cavity and the main residues mentioned during the description should be added to section 2.
  6. The structural models used for the descriptions should be reported throughout the manuscript, not only mentioned in the figure captions. This would help the reader to find the structures used to achieve the structural and mechanistic information reported in the manuscript.
  7. Line 227. “there are” should be removed from the sentence.
  8. Figure 2. The panels showing the structure overlaps are too small and should be enlarged; these panels provide important information and should be highlighted. The color choice for the superimpositions should be improved to better distinguish between the overlapped structures.
  9. Lines 326-328. The identified functionally impairing mutations should be reported.
  10. Lines 376-380. The sentence is not clear, please rephrase.
  11. Line 383. Please remove the parenthesis from “(βH)”.
  12. Line 547. “A human liver protein” is vague, please specify.
  13. Line 583. The reference “(Franzoni et al., 2002)” should be modified in the right reference format.

Author Response

We thank the reviewer for his or her valuable comments and suggestions. Please find below our responses:

1. Only 13 works over the 111 cited in the manuscript have been published in the last 5-years (2016-2021), and only 27 over 111 in last decade (2011-2021).

Response:  We acknowledge the reviewer’s comment and hope that our review will prompt new ideas and development in the field.

2. Figure 1B. The five BABPs used for the structure alignment should be reported in the caption (further their PBD codes). Further the structural model 1tvq, a recent high-resolution crystal structure of clBABP has been reported (PDB code 7O0J).

Response:  We thank the reviewer for bringing the recent structure into our attention. We have included it in Figure 1B. We have also included the name of BABPs in the caption.

3. Citing references should be added after each PDB code throughout the manuscript (and figure captions).

Response:  References after the PDB codes have been included in the revised manuscript in the caption of each figure and throughout the manuscript.

4. Line 52. The hyphen after glycine seems not necessary and could be removed.

Response:  It has been fixed.

5. The description of the binding cavity in BABP complexes, reported in section 6, could be moved before the actual section 2, describing the positive binding cooperativity. A preliminary description of the main features of the binding cavity could facilitate the understanding of the binding cooperativity (section 2), the site preferences (section 3), and the conformational changes occurring upon bile salts binding (section 4). Furthermore, various protein residues reported in these sections are shown in Figure 4, associated with section 6. Introducing Figure 4 before these descriptions would help the reader to have a better idea of the positions of the residues mentioned during the discussion. Otherwise, a figure showing the binding cavity and the main residues mentioned during the description should be added to section 2.

Response:  We agree with the reviewer that a figure showing the key residues mentioned in sections 2 and 3 would help the readability of the manuscript and included a new figure showing superimposed views of the binding cavity of the BABP-complexes discussed in detail in the text. We have also included a sequence alignment of human ileal, chicken ileal, and chicken liver BABPs in panel C of the new figure. Unveiling the full atomic scale description of the binding cavity we preferred doing later in the manuscript but we hope that the new figure highlighting the key differences between the discussed BABP analogues would help the reader understanding the phenomena discussed in the text.

6. The structural models used for the descriptions should be reported throughout the manuscript, not only mentioned in the figure captions. This would help the reader to find the structures used to achieve the structural and mechanistic information reported in the manuscript.

Response:  PDB codes with references have been included throughout the manuscript.

7. Line 227. “there are” should be removed from the sentence.

Response:  It has been fixed.

8. Figure 2. The panels showing the structure overlaps are too small and should be enlarged; these panels provide important information and should be highlighted. The color choice for the superimpositions should be improved to better distinguish between the overlapped structures.

Response:  We have changed the colors and made the panels a bit bigger. With the better choice of color we hope that the differences between the overlapped structures have become more apparent.

9. Lines 326-328. The identified functionally impairing mutations should be reported.

Response:  They have been given in the revised manuscript.

10. Lines 376-380. The sentence is not clear, please rephrase.

Response:  The sentence has been rephrased (pg. 326-331 in the revised manuscript).

11. Line 383. Please remove the parenthesis from “(βH)”.

Response:  It has been fixed.

12. Line 547. “A human liver protein” is vague, please specify.

Response:  It has been clarified.

13. Line 583. The reference “(Franzoni et al., 2002)” should be modified in the right reference format.

Response:  It has been fixed.